# Genome-wide rare variant analysis for thousands of phenotypes in over 70,000 exomes from two cohorts

Elizabeth T. Cirulli [1]*, Simon White [1], Robert W. Read [2,3], Gai Elhanan[2,3], William J. Metcalf[2,3], Francisco Tanudjaja [1], Donna M. Fath[1], Efren Sandoval[1], Magnus Isaksson[1], Karen A. Schlauch[2,3], Joseph J. Grzymski[2,3], James T. Lu[1] & Nicole L. Washington[1]

Understanding the impact of rare variants is essential to understanding human health. We analyze rare (MAF < 0.1%) variants against 4264 phenotypes in 49,960 exome-sequenced individuals from the UK Biobank and 1934 phenotypes (1821 overlapping with UK Biobank) in 21,866 members of the Healthy Nevada Project (HNP) cohort who underwent Exome + sequencing at Helix. After using our rare-variant-tailored methodology to reduce test statistic inflation, we identify 64 statistically significant gene-based associations in our meta-analysis of the two cohorts and 37 for phenotypes available in only one cohort. Singletons make significant contributions to our results, and the vast majority of the associations could not have been identified with a genotyping chip. Our results are available for interactive browsing in a webapp (https://ukb.research.helix.com). This comprehensive analysis illustrates the biological value of large, deeply phenotyped cohorts of unselected populations coupled with NGS data.

---

[1] Helix, 101S Ellsworth Ave Suite 350, San Mateo, CA 94401, USA. [2] Desert Research Institute, 2215 Raggio Pkwy, Reno, NV 89512, USA. [3] Renown Institute of Health Innovation, Reno, NV 89512, USA. *email: liz.cirulli@helix.com

Over the past decade, we have witnessed the growing depth and breadth of genome-wide association studies (GWAS) leveraging genotyped common variants. It has been shown that the most useful and predictive insights about the genetic effects of common variants only begin to appear as sample sizes reach into the hundreds of thousands. Modern resources like the UK Biobank (UKB, www.ukbiobank.ac.uk) that make thousands of phenotypes available to match these genetic data are proving a boon to our understanding of human genetics. In addition to identifying specific variants associated with traits, modern GWAS show that polygenic scores utilizing thousands of common variants together can explain a sizable portion of phenotypic variation and that genetic risk for one phenotype can help explain variation in another[1–3].

Until now, the insights stemming from these large sample sizes have only been available for common and low frequency variants, with comprehensive studies not available below a minor allele frequency (MAF) of about 0.1%. It has been shown that as allele frequencies drop, the effect sizes of associated variants can increase beyond the limits imposed by natural selection on more common variants[4–6]. In rare disease and family-based studies, aggregating phenotypically-similar probands to identify associated groups of rare variants has been crucial to our understanding of disease; exome and genome-based approaches are now standard of care for evaluating these patients[7–10]. However, the impact of rare variants on common traits and sub-clinical phenotypes has only been examined for selected phenotypes as large exome and phenotypic datasets have not been available[11–13].

The release by the UKB of 49,960 exomes matched to thousands of phenotypes enables, for the first time, large-scale clinicalomics discovery, including analyses of rare variants, at scale[14]. We have coupled these data to the exomes of 21,866 participants in the Healthy Nevada Project (HNP, Renown Health, Reno, Nevada) who consented to research involving their electronic medical records[15] (Table 1).

Population-based analyses to identify statistically significant associations between traits and rare variants require a different methodology from the common variant methods to which the field has grown accustomed[16,17]. Because the power to identify statistically significant rare variant associations decreases as the MAF decreases, discovery analyses require that these rare variants be grouped together in some way. The most common method to group rare variants together in population-based genetic analyses is at the level of the gene, usually via a collapsing test (Fig. 1), combined multivariate and collapsing test, the sequence kernel association test (SKAT), or unified tests that can consider both a burden and nonburden situation (SKAT-O)[16,18,19]. Analysis methods that group rare variants have been used in exome and genome sequencing studies to successfully discover genes associated with many traits, such as myocardial infarction, amyotrophic lateral sclerosis, and blood pressure[11,12,20].

Here, we apply a gene-based collapsing analysis method to 49,960 participants for 4264 phenotypes measured by the UKB as well as 1934 traits in an additional 21,866 participants from the HNP cohort. We identify 64 statistically significant gene-based associations in our meta-analysis of the two cohorts, and 37 for phenotypes available in only one cohort. We show the unique power of including rare variants from exome sequence data in analyses by demonstrating the significant contributions of singletons to our results and identifying associations that could not have been discovered with a genotyping chip. Our analysis makes rare-variant discoveries by combining tens of thousands of exomes with thousands of phenotypes across multiple health systems.

## Results

**Description of included datasets**. We used two cohorts in this analysis (Table 1). The first cohort was the set of sequenced UKB exomes. The UKB participants are between the ages of 40 and 69, and each has been extensively phenotyped, including consenting to making their medical records available. As described previously, the exome-sequenced set of UKB samples is enriched for individuals with MRI data, enhanced baseline measurements, hospital episode statistics, and linked primary care records (described for Category 170 at http://biobank.ctsu.ox.ac.uk/crystal/label.cgi?id=170). Of the 49,960 exome-sequenced individuals, 55% are female, and 40,468 are classified by the UKB as genetically of European ancestry (field 22006).

The second cohort included the exomes of 21,866 participants from the HNP (Table 1). These are unselected patients from Northern Nevada (Renown Health, Reno, Nevada) who consented to research involving their electronic medical records[15]. The participants in this cohort are aged 18–89+, and 68% are female. We classified 17,238 of these as European ancestry using principal component analysis based on 184,445 representative common variants (see "Methods").

**Collapsing rare variants**. We performed a gene-based collapsing analysis to identify genes in which rare variants were, in aggregate, associated with a phenotype. In brief, we identified qualifying variants that met specific annotation criteria (see "Methods") and had a MAF < 0.1%. We explored two gene-based collapsing models: (1) all non-benign coding and (2) only loss of function (LoF). The LoF model was used to identify associations where only putative LoF variants had an effect. In the coding model, we included 1,074,012 qualifying variants across 16,341 genes in the UKB cohort and 754,459 variants across 17,023 genes in the HNP cohort (see "Methods"). In the LoF model, we included 165,480 qualifying variants across 15,276 genes in the UKB cohort and 111,735 variants across 14,848 genes in the HNP cohort. There were 15,999 coding model genes and 13,474 LoF model genes that overlapped between the two cohorts. The median number of qualifying coding and LoF variants per gene in the UKB European ancestry population was 34 and six, and 22 coding and four LoF for the HNP cohort, respectively (Fig. 2 and Supplementary Fig. 1). When looking at all ethnicities, these numbers rose to 48 coding and seven LoF for the UKB cohort and

**Table 1 Study and cohort information.**

|  | UK Biobank (UKB) | Healthy Nevada Project (HNP) |
|---|---|---|
| N individuals: total/European ancestry | 49,960/40,468 | 21,866/17,238 |
| N phenotypes: binary/quantitative | 3014/1250 | 1784/150 |
| N phenotypes unique to cohort: binary/quantitative | 1240/1203 | 10/103 |
| Median N cases for binary traits [range] | All: 173 [5:46,192] | All: 153 [5:11,779] |
|  | Eur: 139 [1:37,983] | Eur: 126 [2:9, 426] |
| Median N phenotyped for quantitative traits [range] | All: 10,735 [635:49,904] | All: 2339 [343:19,698] |
|  | Eur: 9287 [516:40,428] | Eur: 1924 [290:15,542] |

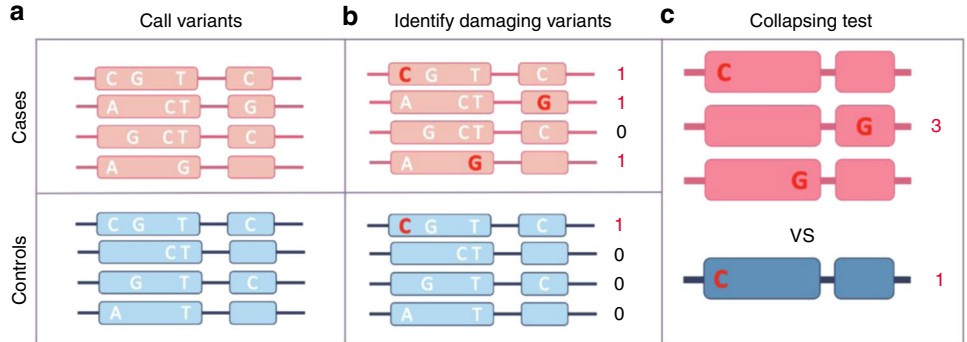

**Fig. 1 Gene-based collapsing analysis. a** First, variants in each gene are identified by sequencing. **b** Variants that are predicted to be damaging—those that are rare and annotated as likely to affect the functionality of the gene, such as coding variants—are then selected for analysis. **c** Finally, the number of cases with a qualifying variant in each gene is compared with the number of controls with a qualifying variant, producing one statistical result per gene instead of one per variant.

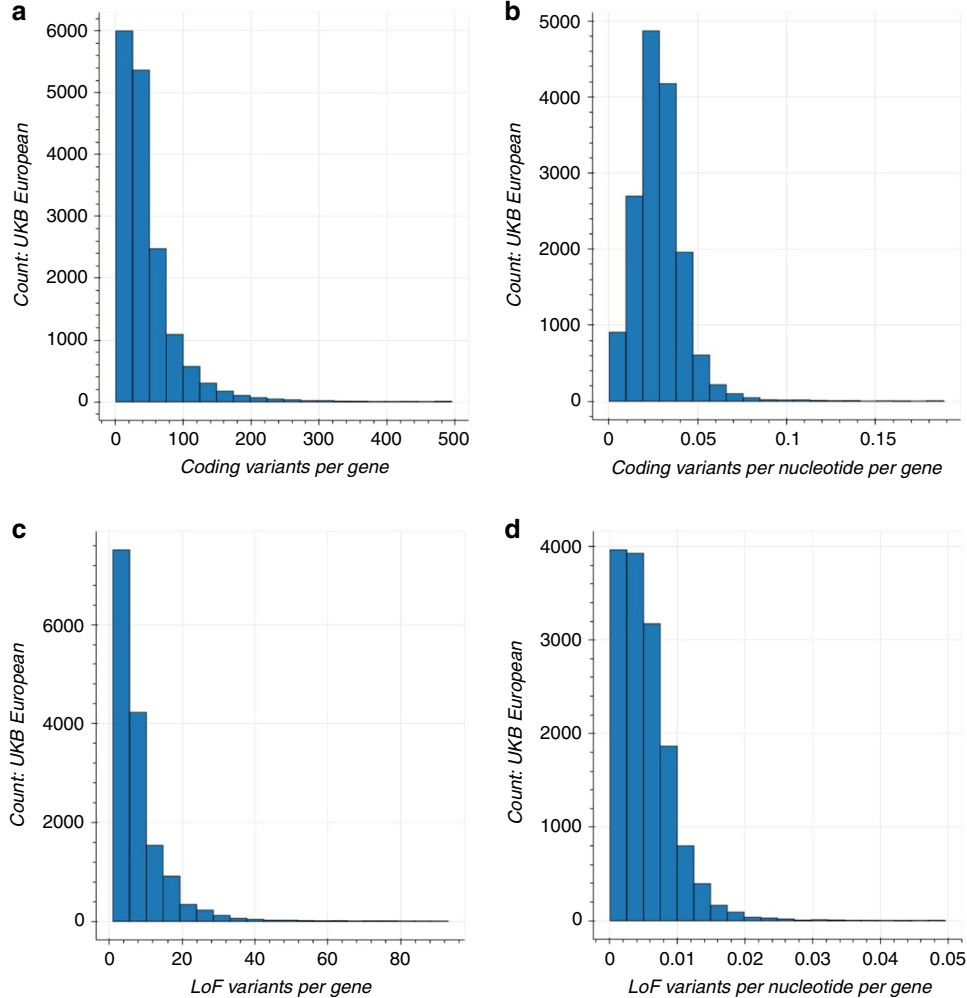

**Fig. 2 Histogram of number of qualifying variants per gene in European UKB cohort. a** Number of qualifying coding variants per gene. Eleven genes with >500 variants were excluded from plot. The median of variants per gene is 34 (range [1:2833]). **b** Number of qualifying coding variants per coding nucleotide of each gene. Sixteen genes with values >0.2 were excluded from the plot. The median of variants per nucleotide is 0.027 (range [0.0001:0.991]). **c** Number of qualifying loss of function (LoF) variants per gene. Six genes with >50 variants were excluded from plot. The median of variants per gene is six (range [1:178]). **d** Number of qualifying LoF variants per coding nucleotide of each gene. Nine genes with values >0.05 were excluded from the plot. The median of variants per nucleotide is 0.005 (range [$9.5 \times 10^{-5}$:0.25]). Plots for all ancestries and HNP cohort can be found in Supplementary Fig. 1.

30 coding and seven LoF for the HNP cohort. The median percentage of people carrying qualifying variants in each gene was the same for both the UKB and HNP cohorts, both in European ancestry and across ethnicities: 0.13% for the coding model, and 0.02% for the LoF model.

**Reducing test statistic inflation.** We performed our main analysis in the European ancestry individuals, including related individuals, using a linear mixed model (LMM) to account for relatedness and population structure (see QQ plots in Supplementary Fig. 2). To reduce test statistic inflation for binary traits, genes were only included in the LMM analysis if the expected number of variant carriers in the case group was at least ten, based on the overall carrier and phenotype frequency[21]. This is an essential step to avoid false positive associations in gene-based collapsing analysis results, especially when there is a case-control imbalance (Fig. 3). This cutoff is for an individual to be carrying any qualifying variant in the gene, and thus it may reflect a situation where ten people are each carrying their own unique variants in a gene, with each variant only being seen once. This criterion does have the consequence of reducing the number of genes that can be investigated, as genes with low numbers of variants will not pass the threshold for all analyzed phenotypes. To compensate and permit coverage of these genes, we performed a supplementary Fisher's exact test analysis of the binary traits that was restricted to unrelated European ancestry individuals. The Fisher's exact test did not require a minimum number of carriers and showed no test statistic inflation (see Fig. 3 and Supplementary Fig. 2). For quantitative traits, we set a minimum threshold of five variant carriers (again, of any qualifying variant

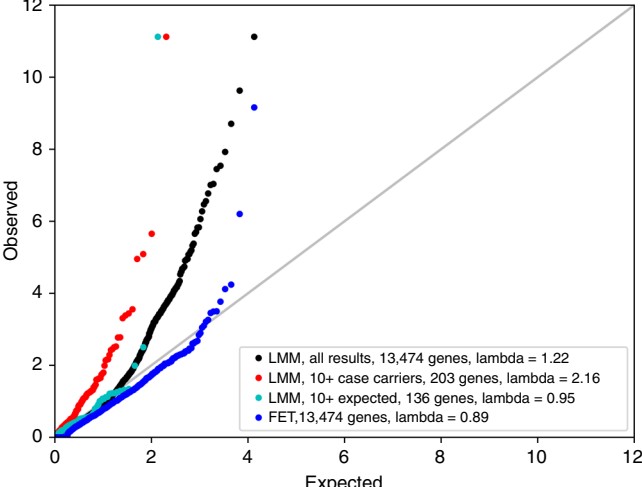

**Fig. 3 Overlaid QQ plots for the coding model with the phenotype atrial fibrillation.** This phenotype has a 1:22 case:control ratio. Shown are the results for a linear mixed model (LMM) meta-analysis of all European ancestry individuals with no minimum number of variant carriers required (black), with at least ten case carriers observed (red), and with at least ten case carriers expected in the case group based on the overall frequency (cyan), as well as a Fisher's exact test (FET) of unrelated European ancestry individuals and all genes included (blue). The second to last condition is the requirement we set for our main analysis results. The one significant association is *TTN*, known from previous studies to be involved in phenotypes related to atrial fibrillation[28]. This association is significant (meta-analysis $p < 3.4 \times 10^{-10}$) in the LMM analysis, but it is difficult to distinguish from test statistic inflation without using the 10 expected case carriers cutoff (cyan). There is no inflation in the Fisher's exact test of unrelated individuals, but this association is not significant in that analysis.

in that gene) to avoid overinterpreting signals coming from very small numbers of carriers.

**Gene-based collapsing meta-analysis.** We analyzed 4264 phenotypes in the currently available European ancestry UKB exome cohort (Table 1, see Supplementary Data 1 for list of phenotypes). The corresponding QQ plots showed a general lack of test statistic inflation (see Supplementary Fig. 2). We then performed the same analyses in the 1934 phenotypes available from the European ancestry individuals in the HNP cohort, which was sequenced at Helix using the Exome+ assay[15]. We next performed a meta-analysis of the results from the two separately analyzed cohorts to identify statistically significant associations with the 1821 phenotypes that had been collected in both cohorts. We identified 47 significant associations (meta-analysis $p < 3.4 \times 10^{-10}$) (Table 2, Supplementary Data 2).

**Incidence of independent replication.** In addition to the meta-analysis, we wanted to clarify how often associations that were statistically significant in the UKB cohort alone would replicate in the independent HNP cohort. The success rate would inform our confidence in associations for phenotypes that were only measured in one cohort and not the other. We therefore also analyzed the European ancestry UKB data as the discovery cohort, with the HNP data as the replication cohort. By this method, we identified 39 associations that were statistically significant (LMM $p < 3.4 \times 10^{-10}$) in the discovery cohort, including 32 of the 47 associations from the meta-analysis and seven that were not significant in the meta-analysis. In the replication cohort, all 39 discovery phenotype/gene combinations showed directions of effect in that were consistent with the discovery signal, 32 (82%) achieved nominal significance (LMM $p < 0.05$), and 18 (46%) achieved statistically significant replication (LMM $p < 0.001$, the Bonferroni threshold for 39 tests). Given that the sample size was much lower in the replication cohort, it is not unexpected that some of the discovery associations did not achieve formal replication significance.

**Analysis of phenotypes unique to each cohort.** The high rate of replication of signals identified in the discovery cohort is encouraging for identifying significant associations for the 2556 phenotypes that could not be incorporated into the meta-analysis: 2443 phenotypes that were measured only in the UKB cohort and 113 that were only in the HNP cohort. This cohort-specific analysis identified 30 additional statistically significant associations, all from the UKB cohort (Table 2, Supplementary Data 2).

**Analysis including all ancestries.** Because we used an LMM, which can account for ancestry differences between individuals, we additionally performed an analysis that included all ancestries. The resulting QQ plots showed a general lack of test statistic inflation (see Supplementary Fig. 2). Of the 77 significant European ancestry associations described above, 64 (83%) generated a lower $p$ value when the additional ancestries were added to the European subset, by a median of 3.4 orders of magnitude. In contrast, for the 17% of associations where the $p$ value increased, it was only by a median of 0.6 orders of magnitude. Given the controlled test statistic inflation for these analyses, analyzing ancestries together in these datasets appears to be a reasonable method to boost power for discovery.

We therefore identified associations that were only significant in the mixed ancestry analysis. We identified 17 from the meta-analysis across cohorts and seven more from the analysis of phenotypes available in only one cohort (Table 3 and Supplementary Data 2). In addition, we found that each of the associations that were statistically significant in the mixed

**Table 2 Statistically significant associations from the European ancestry analysis.**

| Method | Lead gene | Lead phenotype | Lead model | UKB carrier n (%)[b] | UKB p value | HNP carrier n (%)[b] | HNP p value | Meta p value |
|---|---|---|---|---|---|---|---|---|
| LMM | ALPL | Alkaline phosphatase[a] | Coding | 257 (0.68%) | $2.4 \times 10^{-186}$ | 68 (0.65%) | $2.5 \times 10^{-39}$ | $5.3 \times 10^{-223}$ |
| LMM | SLC22A12 | Urate[a] | Coding | 314 (0.83%) | $3.3 \times 10^{-108}$ | 8 (0.46%) | $2.1 \times 10^{-4}$ | $5.5 \times 10^{-111}$ |
| LMM | GOT1 | Aspartate aminotransferase | Coding | 113 (0.3%) | $5.4 \times 10^{-58}$ | 28 (0.27%) | $2.2 \times 10^{-13}$ | $1.6 \times 10^{-69}$ |
| LMM | ABCA1 | HDL cholesterol[a] | Coding | 449 (1.26%) | $2.9 \times 10^{-35}$ | 88 (0.98%) | $2.3 \times 10^{-10}$ | $4.9 \times 10^{-44}$ |
| LMM | APOB | LDL direct[a] | LoF | 96 (0.25%) | $8.9 \times 10^{-31}$ | 8 (0.09%) | $9.7 \times 10^{-11}$ | $8.5 \times 10^{-40}$ |
| LMM | GPT | Alanine aminotransferase | Coding | 126 (0.33%) | $1.9 \times 10^{-28}$ | 40 (0.39%) | $2.4 \times 10^{-6}$ | $4.0 \times 10^{-33}$ |
| FET | HBB | Thalassaemia | LoF | 1 (25%) /0 (0%)[b] | $1.2 \times 10^{-4}$ | 8 (47.06%)/5 (0.03%)[b] | $1.7 \times 10^{-21}$ | $1.2 \times 10^{-24}$ |
| LMM | TUBB1 | Platelet count[a] | Coding | 233 (0.59%) | $1.2 \times 10^{-15}$ | 71 (0.66%) | $2.9 \times 10^{-7}$ | $2.8 \times 10^{-21}$ |
| FET | JAK2 | D45 Polycythaemia vera | Coding | 13 (36.11%)/209 (0.61%)[b] | $1.7 \times 10^{-19}$ | 3 (10.34%)/77 (0.53%)[b] | $5.4 \times 10^{-4}$ | $1.7 \times 10^{-19}$ |
| LMM | KLF1 | Mean corpuscular haemoglobin | LoF | 27 (0.07%) | $5.0 \times 10^{-15}$ | 4 (0.04%) | $8.9 \times 10^{-4}$ | $2.3 \times 10^{-17}$ |
| LMM | APOA5 | Triglycerides | LoF | 42 (0.11%) | $2.0 \times 10^{-12}$ | 10 (0.11%) | $1.1 \times 10^{-2}$ | $1.0 \times 10^{-13}$ |
| LMM | GP9 | Mean platelet volume[a] | Coding | 85 (0.22%) | $5.2 \times 10^{-11}$ | 28 (0.27%) | $1.5 \times 10^{-3}$ | $3.1 \times 10^{-13}$ |
| LMM | ANGPTL3 | Cholesterol | Coding | 175 (0.46%) | $3.2 \times 10^{-10}$ | 67 (0.62%) | $6.8 \times 10^{-4}$ | $8.8 \times 10^{-13}$ |
| LMM | GCK | Glycated haemoglobin | Coding | 58 (0.15%) | $3.0 \times 10^{-12}$ | 9 (0.17%) | $8.4 \times 10^{-2}$ | $9.3 \times 10^{-13}$ |
| LMM | TTN | I48 Atrial fibrillation and flutter | LoF | 41 (2.38%)/311 (0.8%)[b] | $1.1 \times 10^{-11}$ | 12 (1.48%)/132 (0.8%)[b] | $2.8 \times 10^{-2}$ | $7.6 \times 10^{-12}$ |
| FET | COL4A4 | R31 Unspecified haematuria | LoF | 15 (1.2%)/ 51 (0.15%)[b] | $1.1 \times 10^{-8}$ | 5 (0.47%)/7 (0.05%)[b] | $1.0 \times 10^{-3}$ | $1.2 \times 10^{-10}$ |
| FET | BRCA2 | Z40.0 Prophylactic surgery for malignant neoplasm risk-factors | LoF | 7 (12.28%)/154 (0.45%)[b] | $1.1 \times 10^{-8}$ | 2 (9.52%)/ 56 (0.38%)[b] | $3.1 \times 10^{-3}$ | $1.4 \times 10^{-10}$ |
| LMM | CST3 | Cystatin C | Coding | 56 (0.15%) | $9.6 \times 10^{-52}$ | | | |
| LMM | SHBG | SHBG | Coding | 149 (0.42%) | $1.1 \times 10^{-33}$ | | | |
| LMM | LDLR | Non-cancer illness code, self-reported: high cholesterol | Coding | 79 (1.54%)/173 (0.49%)[b] | $1.9 \times 10^{-18}$ | | | |
| LMM | SLC45A2 | Hair colour: Blonde | Coding | 54 (1.16%)/112 (0.31%)[b] | $1.2 \times 10^{-17}$ | | | |
| LMM | STAB1 | Median T2star in putamen (right) | LoF | 38 (0.4%) | $2.1 \times 10^{-14}$ | | | |
| LMM | GP1BB | Mean platelet volume[c] | Coding | 33 (0.08%) | $3.0 \times 10^{-12}$ | | | |
| LMM | SMAD6 | 6 mm weak meridian (right) | LoF | 30 (0.11%) | $3.1 \times 10^{-11}$ | | | |
| LMM | CRP | C-reactive protein | Coding | 66 (0.17%) | $6.6 \times 10^{-11}$ | | | |
| LMM | MC1R | Hair colour: Red | Coding | 31 (1.75%)/222 (0.57%)[b] | $2.2 \times 10^{-10}$ | | | |

D45, I48 and R31 refer to ICD-10-CM diagnosis codes, Median T2star is a measurement from a brain MRI, 6 mm weak meridian (right) is from keratometry of the right eye. When multiple phenotypes and/or models (coding, LoF) were significantly associated with a gene, only the lead phenotype and model are shown. When multiple genes were associated with a trait, only the top gene is shown. All results can be found in Supplementary Data 2
LMM linear mixed model, FET Fisher's exact test, LoF loss of function, HDL high density lipoprotein, LDL low density lipoprotein, SHBG sex hormone binding globulin
[a]Additional genes associated with alkaline phosphatase include GPLD1, ASGR1, and ABCB11; with HDL cholesterol include LCAT, CETP, and SCARB1; with LDL direct include PCSK9; with platelet count include MPL and ITGA2B; with urate include SLC2A9; and with mean platelet volume include IQGAP2, GFI1B, and GP1BA
[b]For binary traits, the information shown is case n (%)/ctrl n (%)
[c]Although this phenotype was included in the meta-analysis, this particular gene did not have carriers in the HNP cohort

**Table 3 Statistically significant associations from the mixed ancestry analysis.**

| Gene | Lead model | Lead phenotype | UKB carrier n (%)[a] | UKB p value | HNP carrier n (%)[a] | HNP p value | Meta p value | Eur carrier n (%)[a,b] | Eur p value[b] |
|---|---|---|---|---|---|---|---|---|---|
| UGT1A1 | Coding | Total bilirubin | 90 (0.19%) | $3.4 \times 10^{-14}$ | 19 (0.15%) | $1.5 \times 10^{-2}$ | $4.3 \times 10^{-15}$ | 77 (0.16%) | $5.5 \times 10^{-9}$ |
| FCGRT | Coding | Albumin | 72 (0.16%) | $4.9 \times 10^{-12}$ | 28 (0.22%) | $2.4 \times 10^{-2}$ | $8.5 \times 10^{-13}$ | 72 (0.16%) | $9.2 \times 10^{-8}$ |
| TMPRSS6 | Coding | Mean corpuscular haemoglobin | 389 (0.8%) | $1.5 \times 10^{-11}$ | 105 (0.78%) | $5.0 \times 10^{-2}$ | $5.8 \times 10^{-12}$ | 369 (0.73%) | $6.3 \times 10^{-9}$ |
| SLCO1B3 | Coding | Total bilirubin | 531 (1.14%) | $6.7 \times 10^{-10}$ | 135 (1.06%) | $1.6 \times 10^{-2}$ | $4.5 \times 10^{-11}$ | 574 (1.18%) | $1.4 \times 10^{-8}$ |
| OCA2 | Coding | Hair colour: Blonde | 32 (0.61%)/65 (0.15%)[a] | $1.5 \times 10^{-14}$ | | | | 31 (0.66%)/46 (0.13%)[a] | $2.5 \times 10^{-15c}$ |
| TYRP1 | Coding | Hair colour: Blonde | 65 (1.24%)/ 231 (0.52%)[a] | $3.4 \times 10^{-12}$ | | | | 55 (1.18%)/180 (0.5%)[a] | $6.6 \times 10^{-9}$ |
| SEC23B | Coding | Red blood cell distribution width | 341 (0.7%) | $1.9 \times 10^{-10}$ | | | | 279 (0.71%) | $3.4 \times 10^{-10}$ |

All results shown are from the LMM with all ethnicities. When multiple phenotypes and/or models (coding, LoF) were significantly associated with a gene, only the lead phenotype and model are shown. All results can be found in Supplementary Data 2
[a]For binary traits, the information shown is case n (%)/ctrl n (%)
[b]Eur carrier and Eur p value columns: For the phenotypes measured in both cohorts, the European meta-analysis values are shown. For the phenotypes measured only in UKB (blank for HNP), the UKB Eur values are shown
[c]While OCA2 was significantly associated with hair colour in the European ancestry subset, that subset had only nine expected case carriers, and so it failed our screening. In the Fisher's exact test in unrelated individuals with no carrier cutoff, the p value is $1.3 \times 10^{-8}$

population also generated $p$ values below $1 \times 10^{-5}$ in the European ancestry subset (with the exception of some associations of *HBB* against various blood phenotypes, as this gene had very few European ancestry variant carriers in the UKB cohort). These associations were therefore already strong in the European ancestry subset but appeared to need a larger sample size to push them to significance.

**Summary of gene-based results**. The vast majority of the significant gene-phenotype associations identified were consistent with the current knowledge in the field (see Supplementary Data 2 for details). For example, rare variants in *PCSK9* and *APOB* were associated with low density lipoprotein (LDL) levels, and rare LoF variants in *TUBB1* were associated with platelet count.

We also found several associations that could be reasonably expected given current knowledge in the field but had not been previously identified in this type of population. For example, we found that rare coding variants in *GP1BB* were associated with higher mean platelet volumes in the general population, consistent with their previous association with some familial bleeding and platelet disorders[22]. As another example, we identified associations between rare coding variants in *TYRP1* and blonde hair. A variant in this gene had previously been shown to cause blonde hair in dark-skinned individuals of Melanesian ancestry from the Solomon Islands, but until now it was thought that this gene did not play a role in blonde hair in other ancestries[23,24].

Additional discoveries were novel. For example, we found that rare coding variants in *STAB1* were associated with median T2star MRI measures in several brain structures, with the strongest association in the putamen. As *STAB1* is a transmembrane receptor that is thought to play a role in angiogenesis, this finding provides novel hypotheses for further study. We also found an association of loss-of-function variants in *SMAD6* with eye measurements. *SMAD6* is a member of the SMAD family of signal transducers, inhibitors of BMP signalling. While its suggested role in human eye development is novel, BMP signalling pathways are involved in many human development processes (reviewed in ref. [25]), including eye development, and a recent study found that the *SMAD6* mouse homologue, Smad6, is essential for blood vessel function in the developing mouse retina[22].

In addition to the associations that met our stringent significance criteria, there were a number of additional associations that are worth mentioning. There were multiple associations that generated significant *p* values in the meta-analysis but did not meet our criteria of having a better *p* value in the meta-analysis than in each individual cohort. For example, we observed a significant association between albumin levels and variants in the gene that encodes albumin, *ALB* (meta-analysis $p = 7.3 \times 10^{-19}$), but the meta-analysis *p* value was higher than the UKB-specific *p* value because there was only one carrier in the HNP cohort. In addition, we identified a novel association between LoF variants in the *PAPPA* gene and decreased height ($p = 1.7 \times 10^{-10}$ in the meta-analysis of unrelated individuals), but there were only two carriers in the HNP cohort and the *p* value rose above significance in the main LMM analysis that included relatives. While this gene has been previously implicated in GWAS of height, this is the first time that rare variants in this gene have been found to be significantly associated with height in a human population[26,27]. As one more example, our meta-analysis identified as significant ($p = 2.5 \times 10^{-20}$) the known association between LoF variants in *TTN* and the ICD10 code I48 for dilated cardiomyopathy[28]. However, there were only three expected case carriers, which failed our LMM threshold, and the Fisher's exact test of unrelated individuals was just shy of significant ($p = 6.9 \times 10^{-10}$). Nonetheless, our main LMM analysis did identify an association between this gene and the related ICD10 code I42 for atrial fibrillation, a more common phenotype that included more expected variant carriers and thus passed our criteria (Table 2, Fig. 3).

**Individual variant analysis**. Mapping the precise effects of each contributing variant can elucidate the underlying biology of an association. We therefore performed association analyses of each individual rare variant to show the effects they had on the overall signal for each associated gene. For example, variants in *SLC2A9* are associated with low urate levels (Fig. 4a, b). The protein encoded by this gene reabsorbs urate in the proximal tubules of the kidneys. Variants that disrupt the transmembrane regions or lower gene expression are known to be associated with hypouricemia[29]. We find that the association signal in this gene is most heavily concentrated in missense variants in the transmembrane regions of the protein, especially in the first half of the protein (Fig. 4a, b). Of the >40 variants associated with decreased urate levels, 88% are in or directly adjacent to a predicted transmembrane region ($p < 0.05$ from a Fisher's exact test comparing the proportion of positively-associated missense variants in or adjacent to this domain compared with outside of this domain).

Likewise, variants in different portions of *GFI1B* have distinct effects on mean platelet volume (Fig. 4c). Consistent with the literature, variants in the zinc finger domains of this gene are associated with increased platelet volumes ($p < 0.05$ from a Fisher's exact test comparing the proportion of positively-associated missense variants in zinc fingers to outside of zinc fingers), but we make the observation that some variants between zinc fingers 3 and 4 may have an effect in the opposite direction ($p < 0.05$ from a Fisher's exact test comparing the proportion of negatively-associated missense variants in this region to outside of this region, even after excluding all zinc finger variants)[30,31].

In one more example, a significant association is observed between variants in *ASGR1* and alkaline phosphatase AP levels. Previously, two LoF variants in this gene were found to be associated with AP levels, coronary artery disease, and non-HDL cholesterol[32]. In our analysis, the association is most strongly influenced by LoF variants, but many missense variants contribute to the signal as well (Fig. 4d).

Finally, we investigated what proportion of the gene-based signals could be traced back to one causal variant as opposed to requiring the grouped effects of multiple rare variants to achieve significance. First, we found that only 16% of the significantly associated genes had a single variant within the gene whose *p* value was statistically significant after multiple test correction. Of these, we found that only 7% of associations had a single variant with a better *p* value than the gene as a whole. Second, we reanalyzed the significant associations after including the lead variant for each gene as a covariate (except for four associations where all qualifying variants in the gene had two or fewer carriers; Supplementary Data 2). After factoring out the signal from the most significant variant in each gene, we found that 41% of associations had their *p* values rise sufficiently to fail statistical significance; yet, 85% of the associations still generated a *p* value below $5 \times 10^{-8}$ with the lead variant accounted for, and only 2% clearly had their entire signal driven by the lead variant ($p > 0.05$ after removing lead variant). Together, our results indicate that while some significant associations had single variants that made major contributions to their effects, few were completely explained by individual variants.

## Discussion

Here we present an analysis to catalogue the effects of rare and unique coding variants on thousands of phenotypes across two large cohorts. Until now, rare variant analyses using next generation sequence data have been performed on a small number of phenotypes at a time such as in schizophrenia, developmental delay, and diabetes[33–35]. Most of these studies were designed around specific phenotypes and collected targeted, disease-specific samples. Analyzing thousands of traits in a biobank-scale population presents additional challenges due to the rarity of the variants, which can lead to false positive associations. The best practices that we suggest to produce reliable results include restricting to high quality regions of the genome, setting a very low MAF cutoff, and requiring that at least a minimum threshold of individuals carry qualifying variants in the analyzed gene. Our

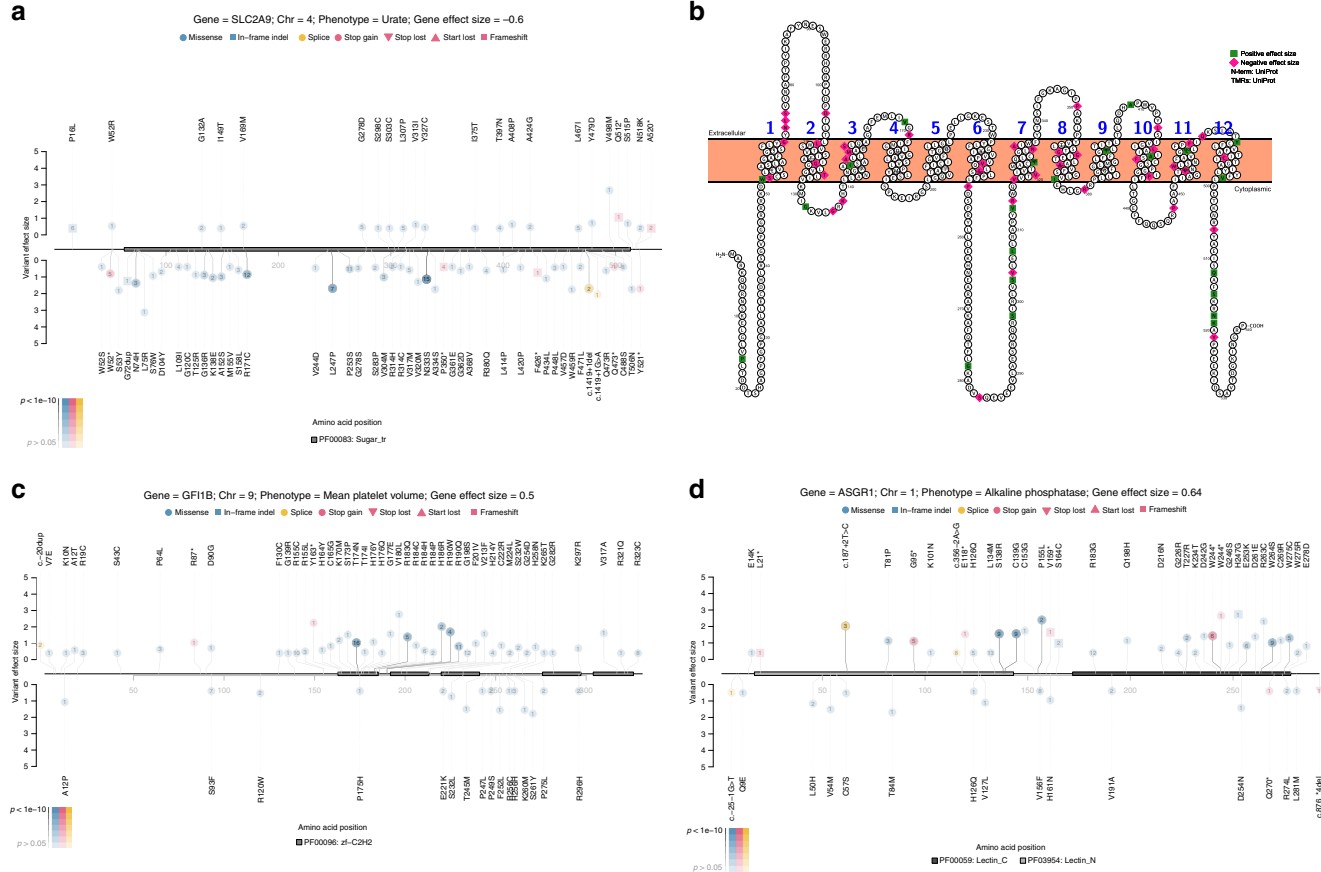

**Fig. 4 Distribution of effects of rare variants in select genes in the UKB cohort. a** SLC2A9 protein and urate levels. The legend shows the gene, its associated phenotype, and the effect size (beta). The effect size is computed from the gene-based collapsing model, in which individuals were coded as either having or not having a qualifying variant. A positive value indicates that variant carriers have, on average, higher values for the phenotype, while a negative value indicates that variant carriers have lower values. The amino acid positions are shown on the x-axis, with the PFAM domain highlighted. The y-axis displays the beta of each individual variant, with negative values shown below and positive values above the horizontal axis. Variants are indicated according to their consequence as shown and labelled according to their amino acid change or splice site variation. The number inside the circle is the number of people carrying that variant. Darker lines connecting the variants to the gene and darker-filled shapes indicate more significant p values for the association. **b** Membrane topology plot of SLC2A9 showing variants with positive effect size (green) on urate levels and variants with negative effect size (pink). SLC2A9 (Glut9) reabsorbs urate in the proximal tubules of the kidneys. Variants that disrupt the transmembrane regions or lower gene expression are known to be associated with hypouricemia[29]. Here, 88% of the variants with negative betas, associated with lowered urate levels, are in or directly adjacent to a predicted transmembrane region, as opposed to only 55% of the variants with positive effect size. **c** GFI1B protein and mean platelet volume. Consistent with the literature, variants in the zinc finger domains are associated with increased platelet volumes, but we make the observation that some variants in between zinc fingers 3 and 4 may be having an effect in the opposite direction[30,31]. **d** ASGR1 protein and alkaline phosphatase levels. In addition to the known effects of LoF variants, we show that missense variants are also playing a role[32]. Plots of the other significantly associated genes are included in Supplementary Fig. 3.

methodology reduces test statistic inflation, and our success is demonstrated by the independent replication of most results.

Our analysis found that the majority of statistically significant gene-based associations were not driven by single explanatory variants. In fact, the signal for these associations was sufficiently dispersed over multiple variants per gene such that if aggregation at the gene-level was not utilized, 84% of the associated genes would not have had a single variant that exceeded the threshold for a multiple-testing correction. Furthermore, when the association for each gene was performed conditional on the single most significant variant in each gene, the p value for the gene as a whole remained below $5 \times 10^{-8}$ for 85% of the associations.

Importantly, the associations identified in this analysis can only be obtained using sequencing techniques, as opposed to chip-based methods. All of the variants used in our analysis have a MAF below 0.1%, which is below the range of frequencies that can currently be comfortably imputed. Furthermore, 38% of the variants included in our analysis were singletons–only observed

once in our dataset and never reported in gnomAD[36]. Such unique variants are not accessible by chip and were vital to our study's success. In fact, 88% of our statistically significant associations generated higher, i.e., worse, p values–by a median of 2 orders of magnitude–when the singletons were removed from the analysis.

In genes where multiple rare variants contribute to the signal, we find that mapping the precise contributions of each variant in the context of the secondary and tertiary structures can reveal the most functional parts of the gene for the given phenotype and provide additional support for a statistical association (Fig. 4). Mapping individual missense variants to their sequence context after a gene-based discovery can also refine the classification of missense variants as true LoF. Corresponding diagrams mapping the locations of the rare variants in each significant gene-phenotype association can be found in Supplementary Fig. 3. Formal statistical tests of domain enrichment and discovery analyses that focus on different gene regions will doubtless

uncover novel associations but will also often have less power due to the small number of people who will be carrying rare variants in each domain.

Our analysis differs from the one presented by Van Hout et al., who performed a gene-based analysis on the European ancestry subset of this same UKB dataset and with some of the same phenotypes[14]. Some of our analysis differences included our use of a replication dataset, our more stringent MAF cutoff (1% vs. 0.1%), and collapsing model differences (LoF vs. both LoF and coding models). We also identified 5 of the 25 gene-based associations reported by Van Hout et al. (between *TUBB1*, *IQGAP2*, and *KLF1* and blood cell phenotypes)[14]. The associations that we did not confirm were largely due to differences in our analysis techniques: the *p* value cutoff (three associations), the MAF cutoff (five associations), the requirement for a minimum number of carriers (six associations, none with *p* values via Fisher's exact test that met our threshold), and restriction to high-confidence regions (three associations) (see Supplementary Table 1 for details). While the reasons for many of these study differences are innocuous, the associations with too few variant carriers in particular can be prone to false signals and will require larger sample sizes to confirm.

Many of the significant signals that we find reflect associations that had already been established in rare familial diseases (see Supplementary Data 2 for details). For example, we show that rare coding variants in *GP1BB* are associated with higher mean platelet volumes; previous studies have shown that variants in this gene cause certain familial bleeding and platelet disorders[22]. Going beyond the acknowledgment that rare variants in this gene can cause these rare conditions, our findings contextualize what it means for any person who carries a non-benign variant in this gene: our work shows more broadly how rare variants in this gene may manifest sub-clinical blood-related phenotypes in the general population. This method brings us closer to a future where a single comprehensive calculation (incorporating both common and rare variation) is able to more accurately predict phenotypic outcomes of polygenic variation towards an improvement in our understanding of human health.

Our analysis has a number of limitations. The analysis included rigid criteria for variant qualification and grouped variants at the most basic level, the gene. Future studies in this dataset can utilize more complex weighting algorithms as opposed to rigid cutoffs and can explore different ways of grouping rare variants, such as by gene family or by exon[18]. Our study used a simple dominant model of inheritance, while recessive models and models that include gene–gene or gene–variant interactions will doubtless provide novel insights as well. Our study was also restricted to the CDS portions of the genome, and future work must expand further, especially to comprehensively analyze rare variants in non-coding regions. Although our initial explorations did not find utility in variance-component score and weighted analysis methods or methods that utilize variants beyond the coding regions (see Supplementary Note on our analyses utilizing the SKAT test and CADD scores[19,37,38]), additional work focusing on these areas will likely identify novel associations, as they have proved useful in some prior studies[39–42]. Finally, many phenotypes included in this study had a small number of cases, which reduced the power for discovery, and will no doubt become better powered as more of these large-scale population studies are completed.

This analysis presents one of the first forays into a new standard for human genetics research. As the sample sizes of cohorts with extensive phenotypic data and next generation sequencing grows, both through publicly available cohorts such as the UKB and population-based screening efforts such as the Healthy Nevada Project, we are now able to investigate the biological impact of rare variants with the same fine-tuned precision with which we currently assess the effects of common variants. A wealth of discoveries await us as we embark on this next phase of incorporating rare genome sequencing information into truly personalized medicine. We provide an interactive browser of our results as a resource to the human genetics community (https://ukb.research.helix.com/) to facilitate these discoveries.

## Methods

**Samples, phenotypes, and variant annotation.** We conducted the UKB research using the UK Biobank Resource under Application Number 40436. We utilized the FE version[43] of the UKB PLINK-formatted exome files (field 23160), a field-standard and stringent method. It has been reported that this version is missing some variants, specifically in regions of the genome with alternate haplotypes, but we proceeded with this version as most genes are unaffected, we restricted to the Genome in a Bottle high-confidence regions of the genome as described below, and a new FE file had not yet been released at the time of this writing[44,45]. We also used the imputed genotypes from GWAS genotyping (field 22801-22823). The HNP study was reviewed and approved by the University of Nevada, Reno Institutional Review Board (IRB, project 956068-12), and all participants provided informed consent. The HNP samples were sequenced at Helix using the Exome+ assay, a proprietary exome that combines a highly performant medical exome with a microarray-equivalent SNP backbone into a single sequencing assay (www.helix.com)[46]. Data were processed using a custom version of Sentieon and aligned to GRCh38, with variant calling and phasing algorithms following GATK best practices[47]. Imputation of common variants in the HNP data was performed by pre-phasing samples and then imputing. Pre-phasing was performed using reference databases, which include the 1000 Genomes Phase 3 data. This was followed by genotype imputation for all 1000 Genomes Phase 3 sites that have genotype quality values <20. Imputation results were then filtered for quality so that only high precision imputed variant calls were reported.

Variant annotation was performed with VEP 95.3[48]. Coding regions were defined according to Gencode version GENCODE 29, and the Ensembl canonical transcript was used to determine variant consequence[49,50]. Variants were restricted to CDS regions. Genotype processing was performed in Hail 0.2.21[51].

For the collapsing analysis, samples were coded as a 1 for each gene if they had a qualifying variant and a 0 otherwise. We defined "qualifying" as coding (stop_lost, missense_variant, start_lost, splice_donor_variant, inframe_deletion, frameshift_variant, splice_acceptor_variant, stop_gained, or inframe_insertion) and not Polyphen or SIFT benign (Polyphen benign is <0.15, SIFT benign is >0.05)[52,53]. We also ran a LoF model that only included LoF variants (stop_lost, start_lost, splice_donor_variant, frameshift_variant, splice_acceptor_variant, or stop_gained). We used a MAF cutoff of 0.1%. To pass the MAF filter, the variant must be below that frequency cutoff in all gnomAD populations[36] as well as locally within each population analyzed: for the UKB exomes, this includes unrelated European, African, East Asian, South Asian, and other ancestry groups. For the HNP exomes, this includes unrelated European, African, East Asian, South Asian, Latino, Native American, and other ancestry groups. For the HNP exomes, only PASS calls were used in the analysis, with an average depth of 62.9×. Qualifying variants were also restricted to the high-confidence regions of the genome as defined by the Genome in a Bottle resource for NA12878[44].

HNP phenotypes were collected from Epic/Clarity EHR data. Microsoft SQL Server was used as a backend database for record storage. SAS 9.4 M5 with SAS/ACCESS to SQL Server was used to perform ETL on these data in preparation for analysis, also in SAS using Base SAS and SAS/STAT. UKB phenotypes were processed using the Neale lab modified version of PHESANT, which rank-transforms quantitative traits to normally distributed data and divides categorical traits into binary sets; HNP quantitative phenotypes were also rank-normalized[54,55]. ICD-10 diagnosis code phenotypes were coded with 1 if participants had the ICD-10 code recorded at least once in their series of Electronic Health Records (EHR), and otherwise with a 0; controls were restricted to one sex when appropriate.

Phenotypes were chosen to have at least 50 cases for binary traits (with at least 5 from each cohort) and 400 phenotyped individuals for quantitative traits across the combined UKB and HNP cohorts. For HNP phenotypes, there was an additional step where the pre-transformed median of quantitative traits was taken when multiple measurements were available for a person.

**Analysis.** We used BOLT-LMM for the main statistical analysis[56]. Briefly, this method builds a LMM using common variants to account for the effects of relatedness and population stratification. The covariates included are age and sex. In the HNP analysis, the Helix bioinformatics pipeline version for alignment and variant calling was also included as a covariate in the model to account for batch effects across different versions of the pipeline.

A representative set of LD-pruned, high-quality common variants were identified for both the creation of principal components and for the random effects and trait heritability in the BOLT-LMM mixed model for collapsed gene analyses and individual variant analyses. Inclusion in this set required MAF >1%, imputation with reasonable accuracy in UKB (information score (INFO) >0.7) and high coverage or imputation in Helix samples (>99% of samples with a sequence-based call or an imputed call with genotype probability (GP) >0.95), LD-pruned

($r^2 < 0.6$) to a set of 184,445 variants. This set of variants was genome-wide, including both coding and noncoding regions.

Our main gene-based LMM analyses required at least five carriers of qualifying variants in analyzed genes for quantitative traits and at least ten carriers of qualifying variants to be expected in the smaller sample group for analyzed genes for each binary trait, similar to previously suggested guidelines[21].

Meta-analysis was performed using the weighted Z-score $p$ value in METAL[57] on the summary stats from each separate BOLT-LMM analysis. We required at least one variant carrier from both the UKB and the HNP groups, the meta $p$ value to be lower (better) than the $p$ values for either individual cohort, and, for binary traits, at least 10 expected case carriers overall for each analyzed gene.

BOLT-LMM determined that 1438 phenotypes had 0 heritability based on the 184,445 common variants in the main European ancestry UKB analysis, and 359 phenotypes in the HNP cohort (for analyses including all ancestries, the phenotype counts were 1287 in the UKB cohort and 82 in the HNP cohort). Analyses of rare variants in these phenotypes therefore could not be completed using the BOLT-LMM algorithm. When quantitative, these phenotypes were analyzed by linear regression of unrelated European-ancestry individuals using PLINK 2.0 with age, sex, and the first 10 European-specific principal components (calculated on these 184,445 variants) included as covariates[58,59]. When binary, these phenotypes were analyzed by Fisher's exact test using PLINK. Fisher's exact test was chosen over logistic regression due to its robustness and lack of test statistic inflation in situations with a small number of case carriers. The Fisher's exact test was also used to identify associations when <10 cases were expected to be carrying qualifying variants based on the overall prevalence. We then performed meta-analysis on the summary stats from each separate PLINK analysis, again using the weighted Z-score $p$ value from METAL[57]. Information on which analyses were performed for which phenotype can be found in Supplementary Data 2.

Information on additional models tried, including a SKAT model and CADD-based cutoffs, can be found in the Supplementary Note.

A conservative Bonferroni correction for multiple tests was set for 17,365 coding model genes (342 unique to UKB, 1024 unique to HNP, and 15,999 overlapping) plus 16,650 LoF model genes (1802 unique to UKB, 1374 unique to HNP, and 13,474 overlapping) × 4377 total phenotypes, thus $3.4 \times 10^{-10}$. For replication analyses, Bonferroni correction was set for the total number of tests done for that particular replication.

We also performed a conditional analysis to test the effects of individual variants on the overall significance of the statistically significant associations. We identified the lead variant contributing to each gene-based signal, i.e., the variant with the best $p$ value per significant gene/phenotype combination. We then performed the gene-based collapsing analysis including that lead variant as a covariate and observed how the $p$ value for the gene as a whole changed (see Supplementary Data 2). For this analysis, we required the lead variant to have at least 3 carriers in the UKB analysis.

Gene plots were made using trackViewer and Protter and annotated with Pfam domains v. 32.0[60–62].

**Reporting summary**. Further information on research design is available in the Nature Research Reporting Summary linked to this article.

## Data availability

UKB data are available for download (https://www.ukbiobank.ac.uk/). Summary statistics for the UKB results are available for download at https://s3.amazonaws.com/helix-research-public/ukbb_exome_analysis_results/README.txt, and also browseable with an interactive web tool at https://ukb.research.helix.com. Summary statistics for the HNP results are available upon request without restriction. Restrictions apply to the availability of the HNP data, which were used under license for the current study, and thus are not publicly available. The HNP data are available for qualified researchers upon reasonable request to Craig.Kugler@dri.edu and Joe.Grzymski@dri.edu and with permission of the Institute for Health Innovation and Helix. Researchers who would like to obtain the raw data related to this study will be presented with a data user agreement, which requires that the participants will not be re-identified and no data will be shared between researchers or uploaded onto public domains.

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

## Acknowledgements

This research has been conducted using the UK Biobank Resource under Application Number 40436. Funding was provided to DRI by the Nevada Governor's Office of Economic Development. Funding was provided to the Renown Institute for Health Innovation by Renown Health and the Renown Health Foundation. We acknowledge A. Buckley and J. Ou for assistance with figures, O. Mendelevitch, G. Sayfan, and T. Michaud for assistance with creating the web resource, and W. Lee and the entire Helix Bioinformatics team for their contributions to the production exome sequencing pipeline. We thank M. Frankovich, T. Curreri, S. Dabe, A. Brain, and all the ambassadors of the Healthy Nevada Project (HNP). We thank Renown Health and DRI marketing for helping to launch the HNP project.

## Author contributions

E.T.C., N.L.W., J.T.L., K.A.S. and J.J.G. designed the study. R.W.R., G.E., D.M.F., E.S. and W.J.M. collected and analyzed the data. E.T.C., N.L.W., J.T.L., K.A.S., J.J.G., S.W., F.T. and M.I. analyzed and interpreted the data. E.T.C. and N.L.W. wrote the manuscript with input from all authors.

## Competing interests

E.T.C., S.W., N.L.W., F.T., D.M.F., E.S., M.I. and J.T.L. are employees of Helix. R.W.R., G.E., W.J.M., K.A.S. and J.J.G. declare no competing interests.
