## [Peer Review File · Nature Communications]

Reviewers' Comments:

Reviewer #1:

Remarks to the Author:

The manuscript describes an exome sequencing association study of 1000's of phenotypes in 50,000 people from the UK Biobank and 18,000 people from the Healthy Nevada Project as replication. Over 100 gene-based associations for a range of phenotypes were identified in the UK Biobank, of which 46% were replicated in the HNP data.

The manuscript is well written, the website will likely be useful and the results are of interest. I do have some concerns though.

1. Having a cut-off of at least 10 carriers per gene is a major limitation. This a concern for two reasons, first this limits quite substantially the number of genes that can be analysed and second there must still be some statistical inflation above the expected 10 carrier threshold. Could a different method be used for the case/control analysis? For example, an exact test would not suffer from inflation. Fisher's exact test is implemented in PLINK which was used here for logistic regression tests. Or could SAIGE be a more appropriate approach than BOLT-LMM as it accounts for extreme case/control bias (Nature Genetics, 2019). Also, I think test statistic inflation should only be a problem for case/control tests - I don't think you should get test statistic inflation for quantitative traits, assuming they have been rank normalised before association testing. I would expect deflation of the test statistics given limited power, if anything.
2. The FE exome sequencing pipeline variants were used for the analysis. An issue with this pipeline was flagged on the UK Biobank genetic analysis mailing list - that 1000's of genes were incorrectly analysed. This was apparently a problem for any gene in a region with an alternative reference haplotype. The authors should comment on this problem and whether it is an issue with their analysis. Also, some justification why the FE pipeline rather than the regerenon called pipeline was used should be provided.
3. More discussion is needed about why some associations reported in Van Hout et al., are not observed in this study. In the discussion it is said to be mainly due to the $MAF < 0.1\%$ and 10 variant carrier requirement. Please provide details of the association that were found in Van Hout et al., but not in this study and the reason why each of them weren't discovered in this study. Are the Van Hout et al. associations false positives?
4. The section in the results "Individual variant analysis" needs some statistical analysis to support some of the conclusions about concentration of variants in different protein domains. While the SLC2A9 variants do seem to be enriched in the transmembrane domains, it's less clear for the other examples.
5. For the LoF analysis were variants in the last exon's excluded? It could be worth including LOFTEE annotations and only including high confidence LoF variants.
6. The use of all individuals in UK Biobank, regardless of ethnicity is still a slight concern to me, especially as I suspect association statistics for very rare variants will not be adjusted as well as common variants by approximate GRM's used by BOLT-LMM. A European-only analysis is provided as a sensitivity analysis, but I wonder whether this shouldn't be the primary analysis and a secondary analysis is the all ethnicities results. Are some of the genes identified by Van Hout et al., but not by this study explained by this difference?

Reviewer #2:

Remarks to the Author:

The manuscript entitled "Genome-wide rare variant analysis of thousands of phenotypes in 68,000 exomes" analyzes rare (MAF<0.1%) variants against 3,166 phenotypes in 49,960 UKBB participants with WES and 1,067 phenotypes in 18,102 exome sequenced members of the Healthy Nevada Project (NHP). In total, 107 gene-based associations with binary or quantitative phenotypes were deemed significant. Of these, the authors replicate 46% in the NHP cohort.

The use of linear mixed models to correct for potential population stratification is a particular strength of the presented approach. The requirement of at least 10 cases expected to carry a qualifying variant is reasonable, but it would be helpful to understand exactly how many genes were analyzed for each phenotype as a consequence of this threshold (e.g. median number of genes and range).

For the replication analysis (Tables 2 and 3), the authors should provide directions of effect along with the exact p-values for each gene in the replication cohort (instead of stating "consistent" or $p < 0.0007$). Presumably all replicated associations with $P < 0.0007$ were direction-consistent with the discovery, but this information needs to be clearly displayed for each gene. For binary traits, please add odds ratio columns.

I did not see Figure 2 being referred to in the text. It would be helpful to provide separate histograms for each of the definitions of qualifying variants used ("non-benign" and "LOF").

Figure 3. Please provide the number of genes included and a genomic inflation factor for each analysis. It would be informative to include the same information for each of the QQ-plots provided in the supplement.

Another gene collapsing model that is of interest would include missense variants that are predicted "damaging" plus LOF. Has this model been considered? If not, please provide the rationale for using only the more inclusive "non-benign" model.

The provision of results in a browsable web app is a strength and would nicely complement supplemental data, but I was not able to access the app remotely over the internet. Please make sure the provided links are operational.

Is HNP exome and phenome data publicly available through dbGAP or other data sharing platform, similar to the UKBB data? Please provide appropriate accession numbers in the methods section. If not available, please provide an explanation in the methods section and in the "Data" section of the "Reporting Summary".

Reviewer #3:

Remarks to the Author:

In this paper, the authors analyse two large WES cohorts in search for rare variant associations. This is a well-written article and a study with unique power, due to its large sample size.

My main comment is that the authors limit their approach to a collapsing test on LoF and non-benign variants, which has important consequences in terms of discovery. It has been shown that p-values/z-scores from tests including LoF or severe variants are only weakly correlated with those arising from an analysis including all coding variants, weighted by a functionality score. This indicates that the authors may discover many more signals if they performed such an analysis.

Furthermore, there is loss of power due to the type of test used. The collapsing test assumes that included variants have a concordant effect size on the phenotype, yet the authors themselves mention in the discussion that some of the observed effects are not concordant.

In my opinion, the paper would benefit greatly from additional analysis using a combined method (e.g. SKAT-O or similar), and a supplementary variant selection method (all exonic + boundary, weighted by functionality score), so as to fully take advantage of this dataset. It may be possible that current methods to do this (weights + SKAT-O + mixed model) are computationally intractable for the sample sizes involved, but the authors do not report having tried to run them.

Further comments:

0. The 4th paragraph where methods are introduced should mention mixed tests such as SKAT-O.

1. Tables 2 and 3 are not main text tables (far too large). They should be moved to supplementary.

2. The way results are reported is not ideal. The authors should first report the results of the meta-analysis, which has the most power to detect signals. Then, results from the traits that do not overlap between the two cohorts should be reported. Single-cohort results with replication information (currently reported in T2 and 3) should be taken out and replaced with meta-analysis results.

3. In the meta-analysis results, the text says : "15 new significant associations (5×10^{-9}) each of which also achieved a nominally significant association (0.05) in [...] the individual cohort". The authors should report p-values in both cohorts in their results table along with the meta-analysis one, and apply an additional filter: the meta-analysis p-value should be smaller than the value for each cohort, otherwise the evidence is actually attenuated by the meta-analysis.

4. The observation that "variants close to the C-terminus may have an effect in the opposite direction" is not particularly meaningful if the p-values associated with these effects are not significant enough. The figure shows that highly significant variants all have positive effects, but it is hard to gauge significance with the currently used alpha scale. I would either clearly indicate insignificant effect sizes or remove this statement. Similarly in figure 4A/B, all significant variants seem to have negative effects, yet 4B does not specifically indicate those. If variants with heterogeneous effect directions were really involved, this would attenuate the p-value given the test used, hence all significant associations will by design be driven by variants in the same direction.

5. "Each of these studies were designed around specific phenotypes and collected targeted, disease-specific samples." There have been rare variant studies for quantitative phenotypes in healthy population cohorts, the authors should mention those (see also comment 7.).

6. "Our analysis found that the vast majority [...] for a multiple-testing correction". A method previously suggested in the literature to properly test for this is to perform conditional analysis on the most strongly associated variant included in the test. If the p-value increases to insignificance or more than a few orders of magnitude, the result effectively reduces to a single-variant test. The more relaxed p-value threshold used in gene-based analyses makes this necessary, in order to make sure that no weak single-variant signals are actually being identified here.

7. The paragraph about the Van Hout et al. study feels like it's been added last-minute. In particular, the text, figures and tables should clearly say which of the signals were previously identified in other rare variant studies (of which Van Hout et al. is only one).

8. The authors do not report precise calculations for their significance thresholds. A rigorous and detailed explanation of threshold calculation should be given in the methods.

Reviewers' comments:

Reviewer #1 (Remarks to the Author):

The manuscript describes an exome sequencing association study of 1000's of phenotypes in 50,000 people from the UK Biobank and 18,000 people from the Healthy Nevada Project as replication. Over 100 gene-based associations for a range of phenotypes were identified in the UK Biobank, of which 46% were replicated in the HNP data.

The manuscript is well written, the website will likely be useful and the results are of interest. I do have some concerns though.

1.1. Having a cut-off of at least 10 carriers per gene is a major limitation. This a concern for two reasons, first this limits quite substantially the number of genes that can be analysed and second there must still be some statistical inflation above the expected 10 carrier threshold. Could a different method be used for the case/control analysis? For example, an exact test would not suffer from inflation. Fisher's exact test is implemented in PLINK which was used here for logistic regression tests. Or could SAIGE be a more appropriate approach than BOLT-LMM as it accounts for extreme case/control bias (Nature Genetics, 2019). Also, I think test statistic inflation should only be a problem for case/control tests - I don't think you should get test statistic inflation for quantitative traits, assuming they have been rank normalised before association testing. I would expect deflation of the test statistics given limited power, if anything.

Response: Thank you for these helpful suggestions. One of our primary interests for this analysis was to include related individuals to boost sample size, which was why we chose the mixed model for analysis. We also did consider SAIGE, but it does not yet work for the scenario of a gene-based collapsing analysis with related individuals included.

However, you are absolutely right that the case/control analyses do not suffer from inflation when performing a Fisher's exact test. As this test must be done on unrelated individuals from a homogeneous ancestry group, we have now supplemented our results with a Fisher's exact test of this subset. This analysis did identify newly significant associations between *JAK2* and D45, Polycythaemia vera, between *HBB* and D56, Thalassaemia, as well as between *BRCA2* and Z40.0, Prophylactic surgery for risk-factors related to malignant neoplasms, as now detailed in the manuscript.

You are also correct that setting the carrier threshold at 10 is not required for the quantitative traits given their rank normalization. However, we also are not comfortable reporting significant associations with genes where only a few people are carriers. We have now set the threshold at 5 carriers, which identified an additional association between LoF variants in *FCGRT* and albumin levels.

1.2. The FE exome sequencing pipeline variants were used for the analysis. An issue with this pipeline was flagged on the UK Biobank genetic analysis mailing list - that 1000's of genes were incorrectly analysed. This was apparently a problem for any gene in a region with an alternative reference haplotype. The authors should comment on this problem and whether it is an issue with their analysis. Also, some justification why the FE pipeline rather than the regeneron called pipeline was used should be provided.

Response: We used the FE pipeline as it represents a stringent, field-standard pipeline, and we have now stated this in the manuscript. It was actually the Regeneron pipeline that had the UK Biobank error, as per their communication on the topic: "a systematic under-marking of duplicate reads has led to errors in the variant-level data. The issue is limited to the exome data processed with the SPB pipeline; those data produced using the FE pipeline are not affected."

1.3. More discussion is needed about why some associations reported in Van Hout et al., are not observed in this study. In the discussion it is said to be mainly due to the $MAF < 0.1\%$ and 10 variant carrier requirement. Please provide details of the association that were found in Van Hout et al., but not in this study and the reason why each of them weren't discovered in this study. Are the Van Hout et al. associations false positives?

Response: We have now included an assessment of why each Van Hout et al. association was or was not replicated in the main text and Supplementary Table 3.

"The associations that we did not confirm were largely due to differences in our analysis techniques: the p-value cutoff (3 associations), the MAF cutoff (5 associations), the requirement for a minimum number of carriers (6 associations, none with p-values via Fisher's exact test that met our threshold), and restriction to high-confidence regions (3 associations) (see Supplement for details). While the reasons for many of these study differences are innocuous, the associations with too few variant carriers in particular can be prone to false signals and will require larger sample sizes to confirm."

1.4. The section in the results "Individual variant analysis" needs some statistical analysis to support some of the conclusions about concentration of variants in different protein domains. While the SLC2A9 variants do seem to be enriched in the transmembrane domains, it's less clear for the other examples.

Response: We have now added p-values to the main text for this section from Fisher's exact tests comparing the number of variants with positive vs. negative associations in each domain as compared to the rest of the gene. The p-values were in the range of 0.005-0.05. See also response 3.6.

1.5. For the LoF analysis were variants in the last exon's excluded? It could be worth including LOFTEE annotations and only including high confidence LoF variants.

Response: Variants were not excluded from the last exon in this analysis. We had also previously performed an analysis that screened variants by LOFTEE annotation, but we found that only 1 of the 41 significant LoF associations from our initial analysis received a better p-value when restricting to high-confidence LoF variants.

1.6. The use of all individuals in UK Biobank, regardless of ethnicity is still a slight concern to me, especially as I suspect association statistics for very rare variants will not be adjusted as well as common variants by approximate GRM's used by BOLT-LMM. A European-only analysis is provided as a sensitivity analysis, but I wonder whether this shouldn't be the primary analysis and a secondary analysis is the all ethnicities results. Are some of the genes identified by Van Hout et al., but not by this study explained by this difference?

Response: We have now restructured the paper to focus on European ancestry as the main analysis. We have also now detailed (see 1.3) the reasons for lack of replication of Van Hout et al. signals, and none were due to the inclusion of diverse ethnicities.

Reviewer #2 (Remarks to the Author):

The manuscript entitled "Genome-wide rare variant analysis of thousands of phenotypes in 68,000 exomes" analyzes rare ($MAF < 0.1\%$) variants against 3,166 phenotypes in 49,960 UKBB participants with WES and 1,067 phenotypes in 18,102 exome sequenced members of the Healthy Nevada Project (NHP). In total, 107 gene-based associations with binary or quantitative phenotypes were deemed significant. Of these, the authors replicate 46% in the NHP cohort.

2.1. The use of linear mixed models to correct for potential population stratification is a particular strength of the presented approach. The requirement of at least 10 cases expected to carry a qualifying variant is reasonable, but it would be helpful to understand exactly how many genes were analyzed for each phenotype as a consequence of this threshold (e.g. median number of genes and range).

Response: As discussed in 1.1, we have now added a supplementary Fisher's exact test for case-control traits and reduced the carrier frequency cutoff for quantitative traits, which increases the number of genes that can be analyzed for each trait. We now add to Supplementary Table 1 information about the number of genes analyzed for each trait.

2.2. For the replication analysis (Tables 2 and 3), the authors should provide directions of effect along with the exact p-values for each gene in the replication cohort (instead of stating "consistent" or $p < 0.0007$). Presumably all replicated associations with $P < 0.0007$ were direction-consistent with the discovery, but this information needs to be clearly displayed for each gene. For binary traits, please add odds ratio columns.

Response: Yes, all directions of effect were consistent in the replicated associations. We have now restructured the manuscript around a meta-analysis instead of the focus on

replication, and we have now included Supplementary Table 2, which provides all of the requested information, including odds ratios.

2.3. I did not see Figure 2 being referred to in the text. It would be helpful to provide separate histograms for each of the definitions of qualifying variants used (“non-benign” and “LOF”).

Response: We now reference Figure 2 and added new histograms as requested.

2.4. Figure 3. Please provide the number of genes included and a genomic inflation factor for each analysis. It would be informative to include the same information for each of the QQ-plots provided in the supplement.

Response: We have added this information to the QQ plots as requested.

2.5. Another gene collapsing model that is of interest would include missense variants that are predicted “damaging” plus LOF. Has this model been considered? If not, please provide the rationale for using only the more inclusive “non-benign” model.

Response: This method is in place to keep from unnecessarily removing all of the variants that do not have Sift or Polyphen annotations, especially indels. In our revised manuscript, we now also include a method that utilizes a CADD cutoff (see response 3.1 and details in the Supplementary Note). As no variants are missing CADD scores, this analysis has the effect of restricting to ones that are predicted to be damaging as you indicate. We found no significant associations by this method that were not already identified in the main coding and LoF analyses.

2.6. The provision of results in a browsable web app is a strength and would nicely complement supplemental data, but I was not able to access the app remotely over the internet. Please make sure the provided links are operational.

Response: We have increased the resources behind our browsable app website and are experiencing fewer outages. Long-term, we plan to migrate our browsable app to a different system, which will be much more reliable and should no longer have access issues. Additionally, the full results are always available for download from https://s3.amazonaws.com/helix-research-public/ukbb_exome_analysis_results/README.txt, and this resource never experiences outages.

2.7. Is HNP exome and phenome data publicly available through dbGAP or other data sharing platform, similar to the UKBB data? Please provide appropriate accession numbers in the methods section. If not available, please provide an explanation in the methods section and in the “Data” section of the “Reporting Summary”.

Response: The HNP datasets are available from the corresponding author on reasonable request, which we have now indicated in the text and reporting summary as requested.

Reviewer #3 (Remarks to the Author):

In this paper, the authors analyse two large WES cohorts in search for rare variant associations. This is a well-written article and a study with unique power, due to its large sample size.

3.1. My main comment is that the authors limit their approach to a collapsing test on LoF and non-benign variants, which has important consequences in terms of discovery. It has been shown that p-values/z-scores from tests including LoF or severe variants are only weakly correlated with those arising from an analysis including all coding variants, weighted by a functionality score. This indicates that the authors may discover many more signals if they performed such an analysis.

Furthermore, there is loss of power due to the type of test used. The collapsing test assumes that included variants have a concordant effect size on the phenotype, yet the authors themselves mention in the discussion that some of the observed effects are not concordant.

In my opinion, the paper would benefit greatly from additional analysis using a combined method (e.g. SKAT-O or similar), and a supplementary variant selection method (all exonic + boundary, weighted by functionality score), so as to fully take advantage of this dataset. It may be possible that current methods to do this (weights + SKAT-O + mixed model) are computationally intractable for the sample sizes involved, but the authors do not report having tried to run them.

Response: We have now performed two SKAT analyses: one that uses the variants from the main analysis but adds beta(1,25) weights based on MAF and one that weights using beta(1,25) and CADD score for all rare coding variants. These analyses were performed on the unrelated European ancestry individuals due to the currently excessive computational requirements for a mixed model SKAT analysis.

We found that only 8% of the significant associations from the SKAT analysis in the European ancestry subset of the UKB cohort were replicated in the HNP cohort. As we have seen in previous works (for example, Long et al. 2017, “Whole-genome sequencing identifies common-to-rare variants associated with human blood metabolites”), when SKAT is applied to real-world data, the results are often noisy and full of significant associations that cannot be replicated. We have added a Supplementary Note to this effect in addition to details on the analysis we performed.

In the spirit of adding association results where variant score is more explicitly taken into account, as well as going beyond coding variants, we also performed gene-based collapsing analyses that were based on CADD scores. The two models we used included all rare variants above the 95% mutation significance cutoff (MSC) for its gene: 1) a restricted model that only included coding variants; and 2) an inclusive model that

collapsed any such variants with 5 kb of the gene, regardless of annotation. We found no new significant associations from these analyses that were not already included in the original coding and LoF results, as we now detail in the Supplementary Note.

Further comments:

3.2. The 4th paragraph where methods are introduced should mention mixed tests such as SKAT-O.

Response: This has now been done.

3.3. Tables 2 and 3 are not main text tables (far too large). They should be moved to supplementary.

Response: We have changed Tables 2 and 3 to summarize the significant associations more succinctly, by pulling out just the top phenotype/model for display. The full results for these significant associations are presented in Supplementary Table 2.

3.4. The way results are reported is not ideal. The authors should first report the results of the meta-analysis, which has the most power to detect signals. Then, results from the traits that do not overlap between the two cohorts should be reported. Single-cohort results with replication information (currently reported in T2 and 3) should be taken out and replaced with meta-analysis results.

Response: We have now restructured the results as suggested.

3.5. In the meta-analysis results, the text says : "15 new significant associations (5×10^{-9}) each of which also achieved a nominally significant association (0.05) in [...] the individual cohort". The authors should report p-values in both cohorts in their results table along with the meta-analysis one, and apply an additional filter: the meta-analysis p-value should be smaller than the value for each cohort, otherwise the evidence is actually attenuated by the meta-analysis.

Response: Our new meta analysis table indicates the p-values for each cohort, and we incorporate the requirement for the best p-value to be from the meta-analysis.

3.6. The observation that "variants close to the C-terminus may have an effect in the opposite direction" is not particularly meaningful if the p-values associated with these effects are not significant enough. The figure shows that highly significant variants all have positive effects, but it is hard to gauge significance with the currently used alpha scale. I would either clearly indicate insignificant effect sizes or remove this statement. Similarly in figure 4A/B, all significant variants seem to have negative effects, yet 4B does not specifically indicate those. If variants with heterogeneous effect directions were really involved, this would attenuate the p-value given the test used, hence all significant associations will by design be driven by variants in the same direction.

Response: It is certainly true that significant associations will only be seen when the majority of the signal in a gene is from variants with effects in the same direction, and the collapsing technique is not intended to pick up signals with effects in different directions. However, some variants can potentially have effects in the opposite direction, so long as their combined effect is not sufficiently strong to wipe out the main effect in the gene.

It is also true that it can be difficult to gauge the effects of single variants when their individual signals are not significant. However, we do feel that meaningful observations can be made from the accumulation of multiple variants, each of which is too rare to drive its own signal, in the same region. We therefore now include p-values from Fisher's exact tests of the overall proportion of variants with positive vs. negative effects in the domains of interest vs. elsewhere in the gene. As shown in the main manuscript text accompanying Figure 4, the p-values for these tests were all below 0.05. This analysis includes all variants analyzed in the genes, regardless of the p-value of their individual effect. However, even removing the variants with the most significant p-values (3 variants for SLC2A9, 4 variants for GFI1B), the overall associations between directions of effect and domain remain.

3.7. "Each of these studies were designed around specific phenotypes and collected targeted, disease-specific samples." There have been rare variant studies for quantitative phenotypes in healthy population cohorts, the authors should mention those (see also comment 9).

Response: We have changed this sentence to begin with "most"; studies of rare variants in healthy cohorts are also mentioned in the introduction.

3.8. "Our analysis found that the vast majority [...] for a multiple-testing correction". A method previously suggested in the literature to properly test for this is to perform conditional analysis on the most strongly associated variant included in the test. If the p-value increases to insignificance or more than a few orders of magnitude, the result effectively reduces to a single-variant test. The more relaxed p-value threshold used in gene-based analyses makes this necessary, in order to make sure that no weak single-variant signals are actually being identified here.

Response: We have now performed this analysis and present the results in the supplement. Our summary of how single variants impacted the analyses, which can be found in the results section, is as follows:

"Finally, we investigated what proportion of the gene-based signals could be traced back to one causal variant as opposed to requiring the grouped effects of multiple rare variants to achieve significance. First, we found that only 16% of the significantly associated genes had a single variant within the gene whose p-value was statistically significant after multiple test correction. Of these, we found that only 7% of associations

had a single variant with a better p-value than the gene as a whole. Second, we re-analyzed the significant associations after taking into account the lead variant for each gene (except for 4 associations where no variant in the gene had at least 3 carriers). After factoring out the signal from the most significant variant in each gene, we found that 41% of associations had their p-values drop below statistical significance: yet, 85% of the associations still generated a p-value below 5×10^{-8} with the lead variant removed, and only 2% clearly had their entire signal driven by the lead variant ($p > 0.05$ after removing lead variant). Together, our results indicate that while some significant associations had single variants that made major contributions to their effects, few were completely explained by individual variants.

3.9. The paragraph about the Van Hout et al. study feels like it's been added last-minute. In particular, the text, figures and tables should clearly say which of the signals were previously identified in other rare variant studies (of which Van Hout et al. is only one).

Response: Supplementary Table 2 is now referenced in the paper and indicates which associations had support from previous studies. The new results section "Summary of gene-based results" highlights some associations and indicates which are already known and which are novel. The legend of Figure 4 indicates which associations in that figure are novel and which are known. We have also now added Supplementary Table 3 to break down the Van Hout et al. replications, and a summary of the comparison with Van Hout et al.'s results to the text as follows (see also response 1.3):

"The associations that we did not confirm were largely due to differences in our analysis techniques: the p-value cutoff (3 associations), the MAF cutoff (5 associations), the requirement for a minimum number of carriers (6 associations, none with p-values via Fisher's exact test that met our threshold), and restriction to high-confidence regions (3 associations) (see Supplement for details)."

3.10. The authors do not report precise calculations for their significance thresholds. A rigorous and detailed explanation of threshold calculation should be given in the methods.

Response: This information is now more clearly stated in the methods.

Reviewers' Comments:

Reviewer #1:

Remarks to the Author:

The authors have done a good job of responding to my comments. There is one outstanding issue though - they state that only the SPB variant calling pipeline has problems. However, both the FE and SPB pipelines have issues. I have pasted below comments from the UK Biobank genetics mailing list which describe the issue with the FE pipeline. There are >1000 genes which are missing variants because of a bug in the FE pipeline. It may be that acknowledging the problem is enough, but it does seem like an important issue to address.

"Hello,

While we are waiting for the new release of SPB exome files, there hasn't been any news on when the FE files will be fixed. The issue about missing variant calls was raised in April (see Mike Weedon's e-mail below) and I haven't seen any replies to this.

I have now dug deeper into the CRAM files themselves, and confirm lack of coverage in certain regions that appear to be those with alternative haplotypes and decoy sequences. I ran Picard metrics on a random set of 155 bam files, and whilst the overall coverage doesn't look too bad (see attached), there are specific stretches of target regions with no coverage. To take the chr17 HNF1B region example from Mike, there is no coverage across a 3Mb region spanning targets within the chr17:35765857-38841389 region. Similarly no coverage in the 180Kb alternative haplotype region spanning chr16:4332281-4514130. That's a lot of genes with zero coverage in FE exomes! I provide a random sample per-target coverage file if anyone wants to check further what's missing.

The second issue I noticed was while doing CRAM->BAM conversion. I used samtools and regardless whether I provided the reference, I got exactly the same BAM. Correct me if I'm wrong, but this means that not all reads from the initial fastq file are in the CRAM files, if I understand CRAM compression correctly those identical to the reference have not been re-created in BAMs?

The samtools command I used is as follows, please let me know if I am not converting correctly:
samtools view -bh -o sample1.bam -T GRCh38_full_analysis_set_plus_decoy_hla.fa sample1.cram

I am also interested in unmapped reads and the converted BAM files have an order of magnitude fewer unmapped reads than some of our in-house exomes. I doubt this is because of an amazingly efficient capture, and again could it be that not all original reads converted to CRAM? Can this be deduced from the attached picard_metrics file? Note I used the xgen_plus_spikein.b38.bed file to specify both baits and targets as I couldn't find a reference to the baits file used for pull-down.

Please can these questions be addressed as a matter of urgency? As it stands the FE release is unfortunately not usable for publishable research.

Many thanks,
Hana

From: UKBiobank Genetics [mailto:UKB-GENETICS@JISMAIL.AC.UK] On Behalf Of Weedon, Michael

Sent: 11 April 2019 14:41

To: UKB-GENETICS@JISMAIL.AC.UK

Subject: Issue with the FE exome pipeline?

Hi,

We've been taking a look at the UK Biobank exome data and comparing the calls from the SPB versus the FE sets. It seems there's 1.7 million more variant calls in the SPB versus the FE set.

The majority of these differences come from about 1000 genes where there are alternate haplotypes. For example, the HNF1B gene has 125 variants called in the SPB set and 1657 in the FE set. Looking at the CRAM files it seems that all the reads mapping to HNF1B have mapping qualities of 0 because the reads also align to an alternate haplotype and presumably that is the reason variants aren't being called in these regions. Note that these alternate haplotypes are present in the headers of the FE cram files but not SPB (e.g. 17_KI270857v1_alt).

We've had a look at approximately 16,000 SNPs which are called by the SPB exome pipeline, are >1% freq and are in HWE ($P > 1 \times 10^{-6}$) and where the genotypes match almost perfectly (genotype concordance >99.9%) with the SNP chip data. Only about 15,000 of these are amongst the FE pipeline calls. And it seems all the missing ones from the FE call set are in these regions where there are alt contigs.

We suspect this is because FE call-set hasn't accounted for the alternate-haplotypes correctly. But we wondered if anyone else had spotted this issue and had another explanation?

Many Thanks,

Mike"

Reviewer #2:

Remarks to the Author:

The authors have adequately addressed my concerns.

Reviewer #3:

Remarks to the Author:

I would like to thank the authors for making substantial changes to the manuscript in response to my and the other reviewers' comments.

I have a few additional comments.

0. 1st paragraph of "Reducing test statistic inflation". Consider rephrasing. E.g. replacing "ancestry differences" with "population structure" (the analysis being described is single-ancestry). Also consider deleting the second sentence, as it feels redundant with the opening statement of "Gene-based collapsing analysis including all ancestries".

1. I suggest removing "further" from the 2nd paragraph of the "Reducing test statistic inflation". As is, it suggests that no filtering was applied for producing Figure S2.

2. Last paragraph before the discussion. Did the authors perform a true conditional analysis, i.e. adding genotypes of the lead variant as a covariate, or simply a leave-one-out analysis? If the latter, it is not sufficient and a true conditional LMM should be performed. I could not find a detailed explanation of this in the methods.

3. Third paragraph of the discussion. Could the authors clarify how exactly singletons were used?

It is mentioned before that the input dataset was filtered for MAC 5 and MAC 10. Singletons by definitions would not pass that threshold?

4. In their response, you mention that "As we have seen in previous works (for example, Long et al. 2017, "Whole-genome sequencing identifies common-to-rare variants associated with human blood metabolites"), when SKAT is applied to real-world data, the results are often noisy and full of significant associations that cannot be replicated." This is likely because SKAT tests a particular type of architecture, just like collapsing tests do. It is not expected that there would be a high cross-replicability between methods. This limitation is the reason why optimal tests were originally proposed. Using SKAT-O as originally suggested would have provided a better subject of discussion, but the authors should nevertheless mention the SKAT results in their discussion, perhaps next to the CADD analysis.

5. before-last paragraph in the Discussion. Studies have actually found variant burdens both using optimal methods, variant weighting and variants beyond coding regions (including regulatory ones) for complex traits. It would be good to add a few references for context.

6. Table 2: replacing E-10 with $\times 10^{-10}$ would improve readability (also relevant for table 3). Some phenotypes such as "6mm weak meridian (right)" should be explained.

Reviewer #1 (Remarks to the Author):

1.1 The authors have done a good job of responding to my comments. There is one outstanding issue though - they state that only the SPB variant calling pipeline has problems. However, both the FE and SPB pipelines have issues. I have pasted below comments from the UK Biobank genetics mailing list which describe the issue with the FE pipeline. There are >1000 genes which are missing variants because of a bug in the FE pipeline. It may be that acknowledging the problem is enough, but it does seem like an important issue to address.

"Hello,

While we are waiting for the new release of SPB exome files, there hasn't been any news on when the FE files will be fixed. The issue about missing variant calls was raised in April (see Mike Weedon's e-mail below) and I haven't seen any replies to this.

I have now dug deeper into the CRAM files themselves, and confirm lack of coverage in certain regions that appear to be those with alternative haplotypes and decoy sequences. I ran Picard metrics on a random set of 155 bam files, and whilst the overall coverage doesn't look too bad (see attached), there are specific stretches of target regions with no coverage. To take the chr17 HNF1B region example from Mike, there is no coverage across a 3Mb region spanning targets within the chr17:35765857-38841389 region. Similarly no coverage in the 180Kb alternative haplotype region spanning chr16:4332281-4514130. That's a lot of genes with zero coverage in FE exomes! I provide a random sample per-target coverage file if anyone wants to check further what's missing.

The second issue I noticed was while doing CRAM->BAM conversion. I used samtools and regardless whether I provided the reference, I got exactly the same BAM. Correct me if I'm wrong, but this means that not all reads from the initial fastq file are in the CRAM files, if I understand CRAM compression correctly those identical to the reference have not been re-created in BAMs?

The samtools command I used is as follows, please let me know if I am not converting correctly:
samtools view -bh -o sample1.bam -T GRCh38_full_analysis_set_plus_decoy_hla.fa
sample1.cram

I am also interested in unmapped reads and the converted BAM files have an order of magnitude fewer unmapped reads than some of our in-house exomes. I doubt this is because of an amazingly efficient capture, and again could it be that not all original reads converted to CRAM? Can this be deduced from the attached picard_metrics file? Note I used the xgen_plus_spikein.b38.bed file to specify both baits and targets as I couldn't find a reference to the baits file used for pull-down.

Please can these questions be addressed as a matter of urgency? As it stands the FE release is unfortunately not usable for publishable research.

Many thanks,
Hana

From: UKBiobank Genetics [mailto:UKB-GENETICS@JISCMAIL.AC.UK] On Behalf Of Weedon, Michael
Sent: 11 April 2019 14:41
To: UKB-GENETICS@JISCMAIL.AC.UK
Subject: Issue with the FE exome pipeline?

Hi,

We've been taking a look at the UK Biobank exome data and comparing the calls from the SPB versus the FE sets. It seems there's 1.7 million more variant calls in the SPB versus the FE set.

The majority of these differences come from about 1000 genes where there are alternate haplotypes. For example, the HNF1B gene has 125 variants called in the SPB set and 1657 in the FE set. Looking at the CRAM files it seems that all the reads mapping to HNF1B have mapping qualities of 0 because the reads also align to an alternate haplotype and presumably that is the reason variants aren't being called in these regions. Note that these alternate haplotypes are present in the headers of the FE cram files but not SPB (e.g. 17_KI270857v1_alt).

We've had a look at approximately 16,000 SNPs which are called by the SPB exome pipeline, are >1% freq and are in HWE ($P > 1 \times 10^{-6}$) and where the genotypes match almost perfectly (genotype concordance >99.9%) with the SNP chip data. Only about 15,000 of these are amongst the FE pipeline calls. And it seems all the missing ones from the FE call set are in these regions where there are alt contigs.

We suspect this is because FE call-set hasn't accounted for the alternate-haplotypes correctly. But we wondered if anyone else had spotted this issue and had another explanation?

Many Thanks,

Mike"

Response: Thank you for bringing this to our attention. We now acknowledge in the manuscript that issues with the FE call-set have been reported, and that the analysis is likely missing some variants. Also, our analysis does restrict to the Genome in a Bottle

high-confidence regions of the genome, which should exclude some of these problematic regions. Our methods now say:

“We utilized the FE version⁴³ of the UKB PLINK-formatted exome files (field 23160), a field-standard and stringent method. It has been reported that this version is missing some variants, specifically in regions of the genome with alternate haplotypes, but we proceeded with this version as most genes are unaffected, we restricted to the Genome in a Bottle high-confidence regions of the genome as described below, and a new FE file had not yet been released at the time of this writing^{44,45}.”

Reviewer #2 (Remarks to the Author):

2.1 The authors have adequately addressed my concerns.

Response: Thank you.

Reviewer #3 (Remarks to the Author):

I would like to thank the authors for making substantial changes to the manuscript in response to my and the other reviewers' comments.

I have a few additional comments.

3.1 1st paragraph of "Reducing test statistic inflation". Consider rephrasing. E.g. replacing "ancestry differences" with "population structure" (the analysis being described is single-ancestry). Also consider deleting the second sentence, as it feels redundant with the opening statement of "Gene-based collapsing analysis including all ancestries".

Response: These changes have now been made.

3.2. I suggest removing "further" from the 2nd paragraph of the "Reducing test statistic inflation". As is, it suggests that no filtering was applied for producing Figure S2.

Response: We have removed this word.

3.3 Last paragraph before the discussion. Did the authors perform a true conditional analysis, i.e. adding genotypes of the lead variant as a covariate, or simply a leave-one-out analysis? If the latter, it is not sufficient and a true conditional LMM should be performed. I could not find a detailed explanation of this in the methods.

Response: Yes, we included the genotypes of the lead variant as a covariate for this analysis. We did *not* perform a leave-one-out analysis. We have now clarified this in the results as follows:

“...we re-analyzed the significant associations after including the lead variant for each gene as a covariate.”

And added the following to the methods:

“We also performed a conditional analysis to test the effects of individual variants on the overall significance of the statistically significant associations. We identified the lead variant contributing to each gene-based signal, i.e., the variant with the best p-value per significant gene/phenotype combination. We then performed the gene-based collapsing analysis including that lead variant as a covariate and observed how the p-value for the gene as a whole changed (see Supplementary Table 2). For this analysis, we required the lead variant to have at least 3 carriers in the UKB analysis.”

3.4 Third paragraph of the discussion. Could the authors clarify how exactly singletons were used? It is mentioned before that the input dataset was filtered for MAC 5 and MAC 10. Singletons by definitions would not pass that threshold?

Response: Our analysis was performed on variants with $MAF < 0.1\%$. The cutoffs of 5 and 10 carriers were for carrying *any* qualifying variant in a gene, not a particular variant. We have added a more clear description of this to the results text:

“This cutoff is for an individual to be carrying any qualifying variant in the gene, and thus it may reflect a situation where 10 people are each carrying their own unique variants in a gene, with each variant only being seen once.”

The analysis that removed singletons thus really did remove the singletons, variants that were seen only once.

3.5 In their response, you mention that "As we have seen in previous works (for example, Long et al. 2017, "Whole-genome sequencing identifies common-to-rare variants associated with human blood metabolites"), when SKAT is applied to real-world data, the results are often noisy and full of significant associations that cannot be replicated." This is likely because SKAT tests a particular type of architecture, just like collapsing tests do. It is not expected that there would be a high cross-replicability between methods. This limitation is the reason why optimal tests were originally proposed. Using SKAT-O as originally suggested would have provided a better subject of discussion, but the authors should nevertheless mention the SKAT results in their discussion, perhaps next to the CADD analysis.

Response: We agree that collapsing tests and SKAT tests are not necessarily expected to identify the same associations, as they do reflect different architectures. However, as detailed in the Supplementary Note, our analysis tested for replication between the UKB and HNP cohorts when the exact same SKAT analysis was run on both of them. Given that the same SKAT test was run in both cohorts, it is discouraging that the rate of replication was only 8%.

SKAT-O performs both SKAT and burden analyses and identifies the best model or combination of models per gene. As we had already performed the collapsing/burden portion of the test, we chose to perform just the SKAT test here.

We have now specifically mentioned the SKAT analysis in the discussion and refer the reader to the Supplementary Note for full details.

“Although our initial explorations did not find utility in variance-component score and weighted analysis methods or methods that utilize variants beyond the coding regions (see Supplementary Note on our analyses utilizing the SKAT test and CADD scores^{19,37,38}), additional work focusing on these areas will likely identify novel associations, as they have proved useful in some prior studies³⁹⁻⁴².”

3.6 before-last paragraph in the Discussion. Studies have actually found variant burdens both using optimal methods, variant weighting and variants beyond coding regions (including regulatory ones) for complex traits. It would be good to add a few references for context.

Response: We have now edited this paragraph to clarify the existence of these studies.

3.7 Table 2: replacing E-10 with $\times 10^{-10}$ would improve readability (also relevant for table 3). Some phenotypes such as "6mm weak meridian (right)" should be explained.

Response: This has now been done.